# Using country-level variables to discover country clusters beyond traditional health policy and performance metrics: An unsupervised machine learning approach for HIV healthcare delivery and financing

Zoïe W. Alexiou[1,2☉], Siddharth Dixit[3,4☉], Osondu Ogbuoji[3], Stefan Kohler[5,6], Rifat Atun[7], Fern Terris-Prestholt[9,10], Iris Semini[9], Caroline A. Bulstra[1,5,7‡*], Till Bärnighausen[5,8,11‡]

1 Department of Public Health, Erasmus MC, Erasmus University Medical Center Rotterdam, Rotterdam, The Netherlands, 2 Department of Health Promotion, Care and Public Health Research Institute (CAPHRI), Maastricht University, Maastricht, The Netherlands, 3 Centre for Policy Impact in Global Health, Duke University, Durham, North Carolina, United States of America, 4 The World Bank, Washington, District of Columbia, United States of America, 5 Heidelberg Institute of Global Health (HIGH), Faculty of Medicine and University Hospital, Heidelberg University, Heidelberg, Germany, 6 Institute of Social Medicine, Epidemiology and Health Economics, Charité– Universitatsmedizin Berlin, Corporate Member of Freie Universitat Berlin and Humboldt-Universitat zu Berlin, Berlin, Germany, 7 Health Systems Innovation Lab, Harvard T.H. Chan School of Public Health, Harvard University, Cambridge, Massachusetts, United States of America, 8 Harvard Center for Population and Development Studies, Harvard University, Cambridge, Massachusetts, United States of America, 9 The Joint United Nations Programme on HIV/AIDS (UNAIDS), Geneva, Switzerland, 10 Warwick Centre for Global Health, Warwick Medical School, Univeristy of Warwick, Coventry, United Kingdom, 11 Africa Health Research Institute (AHRI), KwaZulu-Natal, South Africa

☉ These authors contributed equally to this work.
‡ These authors contributed equally to this work.
* cbulstra@hsph.harvard.edu

## Abstract

HIV remains a major public health challenge, with 1.4 million new infections and 31.6 million people accessing antiretroviral treatment in 2024. In many countries, HIV services have been provided through vertical programs, which, although highly successful in bringing treatment to people living with HIV since the early 2000s, are unlikely to sustain progress beyond donor dependency. The increasing push towards Universal Health Coverage (UHC), while facing reduction in international assistance, are prompting countries search for innovative strategies to effectively deliver HIV services through national systems, supported by domestic financing. Developing country-tailored HIV financing and service delivery approaches will be critical to reaching the end of AIDS as a public health threat and sustaining gains by and beyond 2030. Our study aims to develop an innovative data-driven approach to identify clusters of countries with similar challenges in defining their HIV response sustainability pathways. These clusters provide a framework for mutual learning, allowing countries to co-develop context-specific solutions to HIV financing and

**Data availability statement:** All data used in the study is secondary data from open-access sources. Instructions for access to the data have been provided in the manuscript.

**Funding:** In addition, the financial statement should be updated the financial statement to: "This work was supported by the Joint United Nations Programme on HIV/AIDS (UNAIDS) (project number PR2020/1050056 to CAB; project number PR2020/1050056 to TB) and the Dutch Research Council (NWO) Rubicon program (project number 452022313 to CAB). The funders had no role in study design, data collection and analysis, decision to publish, or preparation of the manuscript.

**Competing interests:** The authors have declared that no competing interests exist.

service delivery. We utilized unsupervised machine learning (ML) methods, including partitional, hierarchical, spectral, and density-based algorithms, to identify clusters of countries based on HIV epidemic and response data among 134 LMICs. We pooled open-source data from repositories covering indicators related to HIV epidemic and response, UHC commitment and progress, legislation surrounding human rights and HIV response, wealth and equity. We identified four country clusters, which did not align with conventional global regions but instead cut across them, revealing more nuanced groupings. Clusters were located in (1) South Asia, East Africa, and Oceania; (2) Sub-Saharan Africa and the Caribbean; (3) Eastern Europe, Middle East, Latin and Southern Africa; (4) Latin America, Asia, Middle East, North Africa, Oceania and the Caribbean. Our study is an early example of how ML techniques can be applied to health policy and (public) health performance.

## Introduction

HIV continues to be one of the most significant public health challenges, with 1.4 million new infections and 31.6 million people accessing antiretroviral treatment in 2024 [1]. In many low- and middle-income countries (LMICs), HIV services are delivered through donor-funded vertical programs. These programs have enabled a rapid scale-up of HIV prevention and treatment – saving millions of lives, particularly in sub-Saharan Africa [2,3]. However, as disease burdens evolve and non-communicable diseases such as diabetes, hypertension, and depression rise, health systems must adapt to deliver a broader range of services [4,5]. At the same time, roughly 400 million people worldwide still lack access to essential health services or face catastrophic health costs, highlighting the urgency of achieving Universal Health Coverage (UHC) as part of the Sustainable Development Goals (SDGs) [6]. A reorientation of health systems towards investment in primary health care, offering a broad range of essential health services, will be required [4].

Despite remarkable progress, projections suggest that the number of people living with HIV will continue to increase until at least 2039, underscoring the need for sustainable, country-led systems capable of providing lifelong care [7]. Yet, recent cuts in U.S. global health funding and international assistance, including President's Emergency Plan for AIDS Relief (PEPFAR) support, threaten to undermine both HIV programs and broader health system transformation across LMICs [8]. Many people living with HIV cannot access life-saving services due to pervasive stigma, inequity in health care, sudden funding reductions, and emerging health threats [9]. Achieving ambitious global goals will therefore require reimagining the HIV response—shifting from donor-driven, vertical programs to sustainable, nationally owned systems that deliver people-centred, affordable services financed through domestic mechanisms [5,6].

To date, over 30 countries have initiated developing pathways to transition the HIV programmes to sustained HIV responses, delivered through national systems, domestically-financed, and politically supported self-reliance [10,11]. However, the

implementation of such mechanisms is fractured, characterized by different programs, funding sources, and phases across various countries [12].

Progress towards sustaining HIV response impact, however, will require context-specific strategies shaped by political leadership, service quality, health care system capacity, supportive policies, and available financing [10,13]. The pace and type of solutions will vary based on how factors across these five areas interact within each country. Effective and equitable delivery of the HIV response through national health care systems depends on overall system performance, primary health care coverage, streamlined integrated delivery models, and person-centred care [12]. Increasing domestic financing will depend on countries' fiscal capacity, political commitment to HIV and health investments, and competing national priorities. Stigma and human rights violations, including against key and vulnerable populations at risk of or infected with HIV, perpetuate inequitable service access, driving new infections to increase and poor viral suppression rates [9].

Given these differences, there is a growing need for a more nuanced understanding of where countries stand in their transition toward sustainable, nationally led HIV responses. Currently, policymaking is largely done by region, and based on epidemic burden and available funding sources, which overlooks the fact that countries within the same region may exhibit distinct epidemic and systemic characteristics [14,15]. Sustainable financing for the HIV response requires customized approaches—"one size fits all" will not suffice. For example, while some countries prioritize integrating HIV services into health insurance benefit packages, others focus on strengthening primary care infrastructure. Identifying clusters of countries with similar challenges can help foster South-to-South learning and guide strategic investments to sustain epidemic control.

Machine learning (ML) methods are increasingly utilized in global health and policy domains for their ability to identify patterns in complex data, enabling more effective, context-specific interventions at macro level. This includes the use of unsupervised ML to update existing typologies to describe and categorize increasingly difficult to describe health systems [16–20]. This study applies unsupervised ML to cluster countries using selected multi-criteria critical to enabling sustained epidemic control through national systems beyond donor dependency. The objectives are to: (1) identify distinct country clusters; (2) assess their robustness; and (3) interpret these clusters through key contributing variables and comparison with existing typologies.

## Methods

### Data

**Data sources and indicators.** We pooled open-source country-level data from various data repositories for all 134 LMICs. We defined domains that could impact sustainable HIV service delivery and financing, as shown in **Table 1**. These domains were derived from UNAIDS' framework for achieving sustainable HIV response results and corresponding criteria to guide country actions (**Text A in S1 Text**) [21]. The UNAIDS multi-criteria framework to guide country actions includes areas on rights and equity Status, ability to pay, disease Burden, pace of progress towards control of the epidemic, and health care system robustness, measured through progress in UHC. We included indicators that represent these areas in the UNAIDS framework and could facilitate the diagnosis of a country's key challenges and help group countries according to their contexts and needs. The selected indicators were across (i) healthcare, Universal Health Coverage commitments and progress, (ii) the HIV/AIDS epidemic, (iii) HIV/AIDS response: healthcare and prevention, (iv) healthcare financing, including for HIV/AIDS, and (v) legislation about human rights and the HIV response. We chose equity indicators, such as disease burdens among vulnerable populations and wealth inequalities, in line with the SDG's and the UNAIDS global AIDS strategy 2021–2026 [22,23]. In addition, we included general country-level information related to (vi) demographic and socioeconomic indicators, (vii) other disease burdens, and (vii) overall health and mortality. We created a cross-sectional database with 120 variables. Details of various variables included in each domain can be found in **Table A in S1 Text.**

**Table 1. Overview of the extracted data by subgroup and source.**

| Subject domain | Extracted variables | Sources |
|---|---|---|
| Healthcare system, Universal Health Coverage commitments and progress | UHC service coverage index; hospital beds[a]; community health workers[a]; nurses[a]; dentists[a]; pharmacists[a]; vaccination coverage children; unmet need for family planning; tuberculosis detection rate; universal health insurance schemes | Demographic and Health Surveys (DHS); UNICEF; WHO |
| HIV/AIDS epidemic | PLHIV; HIV prevalence; new HIV infections; HIV prevalence sex workers; HIV prevalence MSM; HIV prevalence transgender people; HIV prevalence PWID; HIV prevalence prisoners | Demographic and Health Surveys (DHS); WHO; UNAIDS AIDS Info |
| HIV/AIDS response | ART coverage; ART for PMTCT; ART distribution via community health workers; presence national HIV prevention plan; 90-90-90 commitments and progress; facilities delivering integrated HIV services (per 10,000 population) | UNAIDS; National Commitments and Policy Instrument (NCPI) |
| Healthcare financing, including for HIV/AIDS | GDP growth; healthcare expenditure as % of GDP; government health expenditure; external resources for health; out-of-pocket expenditure; catastrophic spending on health; HIV expenditures key populations; Global Fund resources; total spending on HIV/AIDS; COVID-19 Debt Service Suspension Eligibility | WHO; UNAIDS Resources and Financing; World Inequality Database |
| Legislation about human rights and the HIV response | CPI score; political stability; HIV surveillance system in place; routine user fees charges; compulsory detention; criminalization of key risk groups for HIV; human right monitor enforcement; national strategy against gender-based violence; restrictions to civil society regarding HIV service delivery | WDI-World Bank; National Commitments and Policy Instrument (NCPI) |
| Demographic and socio-economic indicators | per capita GDP; age dependency ratio; total per capita wealth; urban population (as % of total); access to basic sanitation | WDI (World Development Indicators)-World Bank, United Nations Population Division. |
| Health and mortality | Life expectancy; Maternal mortality ratio; under-5 mortality; tobacco use prevalence; diabetes prevalence; COVID-19 case fatality ratio | WHO Global Health Observatory; Demographic and Health Surveys (DHS); International Diabetes Federation |

Abbreviations: PLHIV = people living with HIV, PWID = people who inject drugs, ART = antiretroviral therapy, PMTCT = prevention of mother-to-child transmission.

**Data processing.** If available, we included the estimate for each variable from 2017 or later. If such current estimates were not available, we expanded the time window to include data back until 2000 to be able to use most of the pre-selected relevant variables. We transformed some variables into ratios (e.g., people using at least basic sanitation services (% of population): Q1/Q5) to take relative differences into account. Variables with data availability in less than 75% of countries were dropped. We used the k-nearest neighbour algorithm for the remaining variables to impute the missing information [24]. Details on the strategies adopted to manage the data missingness issue can be found in **Table A in S1 Text**.

## Data analysis

We used unsupervised ML algorithms to cluster country-level data, using several approaches including partitional (Partitioning Around Medoids; PAM), hierarchical, spectral, and density-based (DBSCAN) methods [25]. As a pre-processing step, Gower distances were computed to handle mixed data types, followed by Uniform Manifold Approximation and Projection (UMAP), a non-linear dimensionality reduction technique using default parameters (15 neighbours and a minimum distance of 0.1) [26].

PAM clustering was applied using the Euclidean distance metric, with default initialization and iterative optimization until convergence. We used the Silhouette Index (with values above 0.50 considered acceptable based on reference standard) and the Gap Statistic (with an substantial increase indicating better clustering) to assess the optimal number of clusters for PAM, iterating between solutions with 2:5 clusters [13]. Cluster stability was evaluated using bootstrapping with mean Jaccard bootstrap values across 1000 samples (>0.70 considered stable).

Hierarchical (HC) and spectral clustering were applied specifying four clusters, and DBSCAN was run with eps = 0.5 and minPts = 5, resulting in 4 clusters and no points unclustered. The Adjusted Rand Index (ARI) was used to compare consistency across all four clustering algorithms. An ARI above 0.70 indicates strong consistency, 0.50–0.70 suggests moderate agreement, and below 0.50 indicates poor consistency [14,15]. We then used silhouette plots, 2D UMAP plots and country maps to visualize country typologies across algorithms.

The most robust and easy-to-interpret cluster solution was then chosen and used for downstream analyses to aid interpretation of clusters. Fuzzy clustering was additionally used to examine membership probabilities at cluster boundaries, which were visualized using country maps [20].

Random Forest, an ensemble of decision trees, was applied post-hoc to identify the variables most important for distinguishing clusters, out of the box error (OOB) was used to estimate the prediction error. Variable importance was quantified using MeanDecreaseAccuracy, which measures how much model accuracy decreases when a variable is excluded, and MeanDecreaseGini, which reflects how much a variable contributes to reducing node impurity across the trees [27]. The top 20 key input indicators were then selected based on the Random Forest results. We calculated correlation coefficients (Cramer's V Coefficient ($\varphi c$)) to assess if clusters represent groupings that are recognizably different from existing typologies by world region and HIV/AIDS epidemic type.

Finally, each cluster was summarized into a policy-interpretable "profile name" by extracting original mean or median values of key variables identified by the Random Forest. These profiles were then refined by consensus within our multidisciplinary team, incorporating both HIV/AIDS prevalence and region alongside new insights from this study.

We used R version 4.1.2 (The R Foundation, Vienna, Austria) and RStudio version 2022.02.2 + 485 (RStudio, Boston, Massachusetts) for data pre-processing and analysis. The complete code for the analysis can be found on GitHub [28].

## Results

Using PAM clustering, we found four clusters (k = 4) solution robust and interpretable, as the silhouette score remained above 0.50, with acceptable scores for k = 2 (0.671), k = 3 (0.583), and k = 4 (0.527), coupled with a favourable visual assessment of the Gap Statistic metrics (**Fig 1**; Fig A in **S1 Text**). Bootstrap validation showed the following mean Jaccard bootstrap values: cluster 1 (0.779), cluster 2 (0.928), cluster 3 (0.934), and cluster 4 (0.887).

Comparison between the four clustering algorithms showed that PAM and hierarchical clustering (HC) had very similar average silhouette scores (0.53 and 0.53, respectively), indicating good cluster separation, while spectral (0.46) and DBSCAN (0.44) showed slightly lower, but still acceptable, silhouette values. Adjusted Rand Index (ARI) values demonstrated strong agreement between PAM and HC (0.929), as well as between PAM and spectral clustering (0.835). In contrast, agreement with DBSCAN was lower, particularly with PAM (0.710) and HC (0.672), indicating moderate consistency (**Table B in S1 Text**; **Fig B in S1 Text**).

Visual inspection of 2D UMAP plots shows that PAM and HC produce well-separated clusters with clear gaps between groups. Spectral clustering yields similar groupings but with slightly more overlap, while DBSCAN identifies dense regions with moderate consistency and some overlap (**Fig 2**).

Given that PAM clustering was the most robust solution and corresponded closely to other algorithms, we further examined the geographical distribution of the clusters in **Fig 3**. The clusters did not correspond strictly to global regions but cut across them. Cluster 1 (blue) included countries in South Asia (e.g., Afghanistan, Pakistan, Nepal, Bangladesh), East Africa (e.g., Eritrea, Somalia, Tanzania, Sudan), and a few from Oceania (e.g., Myanmar, Cambodia). Cluster 2 (green) was dominated by Sub-Saharan Africa (e.g., Nigeria, Kenya, Zambia, Zimbabwe, South Africa) along with Haiti and Jamaica. Cluster 3 (yellow) brought together a wide geographical range, including Eastern Europe (e.g., Ukraine, Belarus, Serbia), the Middle East and North Africa (e.g., Egypt, Jordan, Algeria), Latin America (e.g., Brazil, Mexico, Argentina), and several Asian countries (e.g., China, Iran, Turkey). Cluster 4 (orange) combined Latin American countries (e.g., Peru, Venezuela, Dominican Republic), Asian countries (e.g., India, Mongolia, Indonesia), North African and Middle Eastern

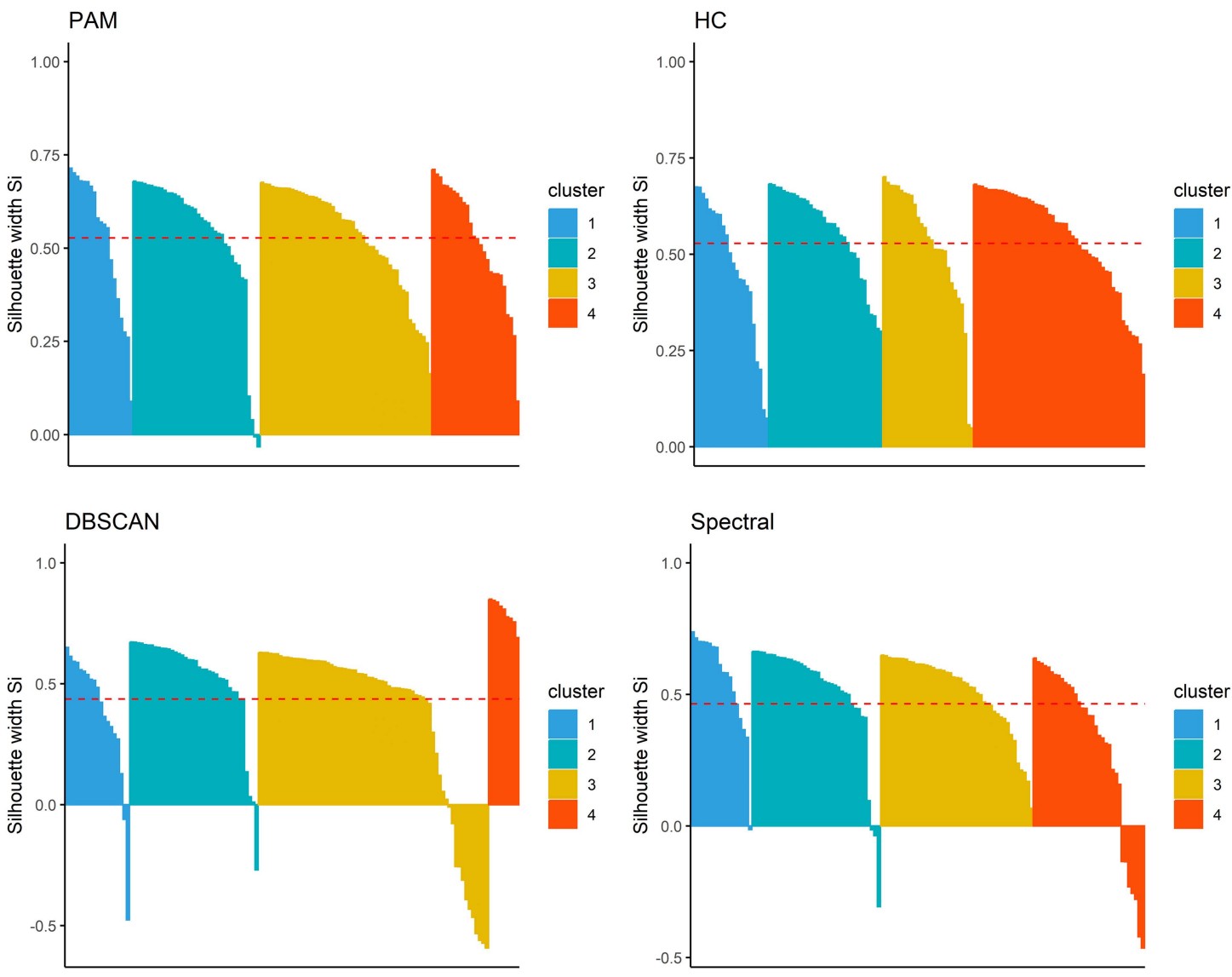

**Fig 1. Silhouette scores for each clustering algorithm.** The silhouette score represents how well an observation fits within its assigned cluster compared to neighbouring clusters, with higher values indicating better-defined and more cohesive clusters. PAM = partitioning around medoids, HC = hierarchical, DBSCAN = density-based spatial clustering of applications with noise.

countries (e.g., Iraq, Morocco), and multiple small island states in Oceania and the Caribbean (e.g., Kiribati, Samoa, Tonga, Grenada). Country assignments for the other clustering algorithms can be found in **Tables C-F in S1 Text**.

**Fig 4** shows countries colour-coded by the degree of belonging to each cluster based on fuzzy PAM clustering, with darker colours representing a higher likelihood of belonging to the cluster. It showed that most countries had a high probability of belonging to a single cluster, such as Mozambique (81.5% cluster 2) and Sri Lanka (75.8% cluster 3). Others, however, displayed more mixed memberships, for example Democratic Republic of Congo (41.1% in cluster 1, 43.1% in cluster 2) (full distributions can be found in **Table G in S1 Text**). This indicates overlap between typologies and highlights that discrete clustering may underestimate uncertainty at cluster boundaries as displayed in **Fig 3**.

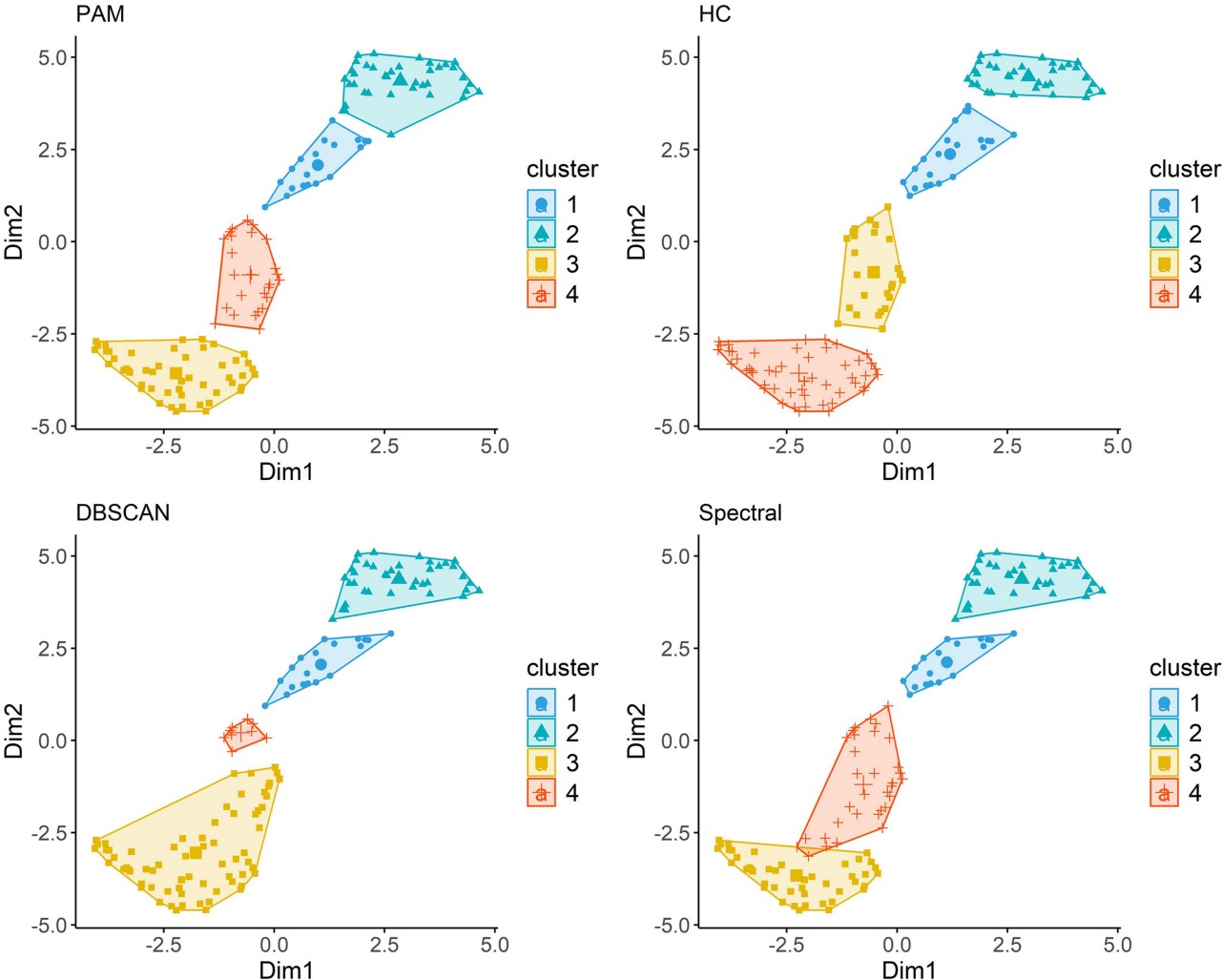

**Fig 2. Two-dimensional UMAP plots.** Each point represents a country positioned along the first and second UMAP dimensions. The spatial distribution reflects similarity across all input indicators, with clusters appearing as groups of points that are closer together in this reduced-dimensional space. PAM = partitioning around medoids, HC = hierarchical, DBSCAN = density-based spatial clustering of applications with noise.

The Random Forest classifier achieved good performance in predicting the PAM clusters (OOB error 11%). Cluster 3 was classified perfectly, while clusters 1 and 4 showed some misclassification, mainly with cluster 2 (**Tables H and I in S1 Text**). The Random Forest classifier also identified the key variables driving cluster assignment. The most influential variables spanned the subject domains Universal Health Coverage (dentists density, physician density, nurse-initiated ART access), HIV/AIDS epidemic (adult HIV prevalence), healthcare financing (international and public funding ratios for HIV, e.g., for prevention, treatment and vertical transmission, Global Fund resources), demographic and socioeconomic factors (population age structure, access to basic sanitation, per capita GDP), and health and mortality outcomes (life expectancy at birth and age 60, maternal and under-five mortality rates) (**Table 2**, **Table J in S1 Text**). Indicators from the subject domains *'HIV response'* and *'Legislation about human rights/the HIV response'* were not included among the top 20 variables but showed measurable contributions to cluster assignment.

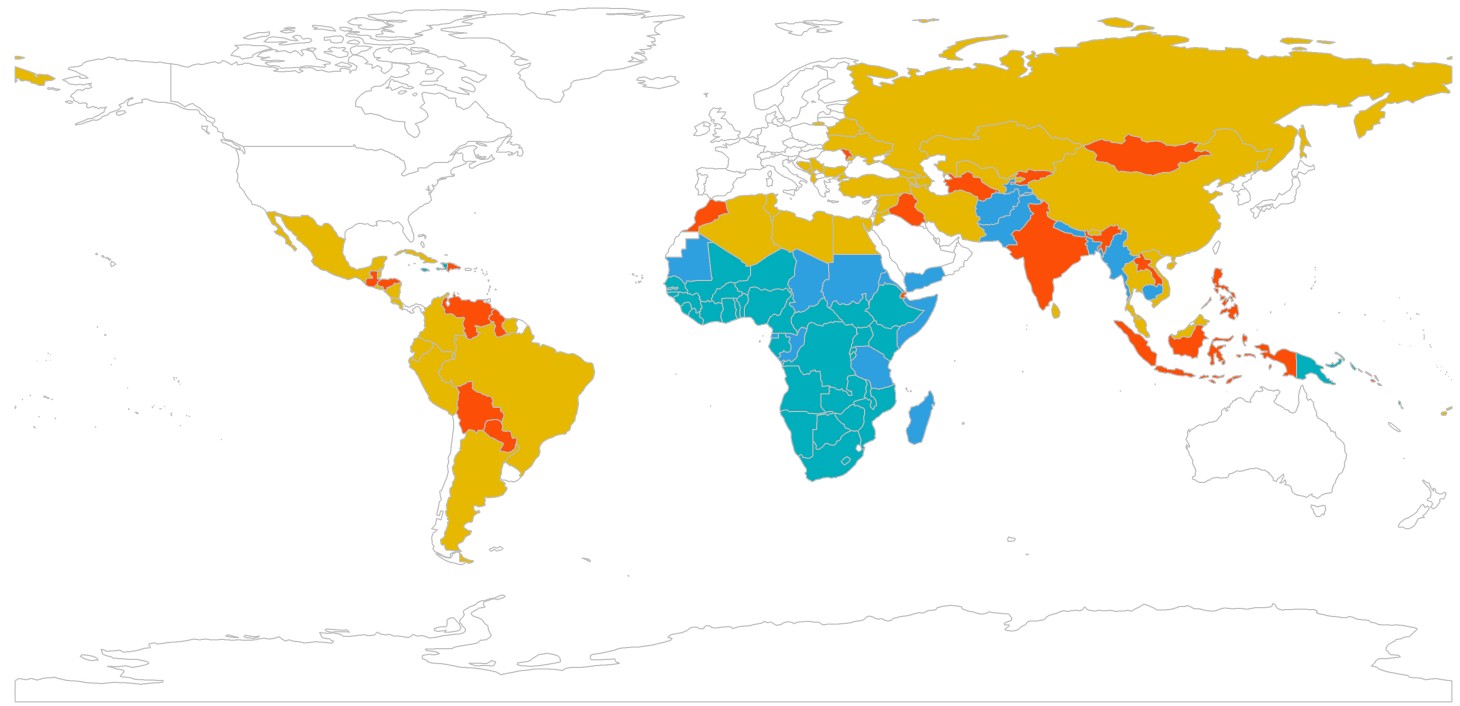

**Fig 3. Country maps for Partitioning Around Medoids cluster assignments.** Each country is assigned to one of the identified clusters based on similarities across indicators. The map visualizes how countries group together according to these patterns. Source of the basemap shapefile: a Spatial-PolygonsDataFrame object containing a simplified world map with polygons attributed to 244 countries, based on Natural Earth data (© Natural Earth, www.naturalearthdata.com), public domain.

The correlation between identified groupings and world region ($\varphi c = 0.54$) and HIV/AIDS prevalence indicators in general and key populations ($\varphi c = 0.15$- 0.46) were low to moderate (Table 2).

## Discussion

Applying unsupervised ML in this context is non-trivial due to the difficulty of integrating different and multi-dimensional data sources. Using a complex and heterogeneous dataset, our study successfully extracted distinct clusters using unsupervised ML algorithms, which offer an alternative to traditional country-clustering by world region or HIV disease prevalence and incidence. Based on multidimensional open-source data, we identified four country profiles of HIV service delivery and financing. Our post-hoc analysis showed that countries cluster primarily according to HIV prevention and treatment coverage, health system resources (e.g., GDP per capita, Global Fund support, and UHC service coverage), service delivery capacity (e.g., health workforce, access to basic sanitation), and demographic structure, and are notably different from existing typologies based solely on HIV/AIDS prevalence and country region.

The analysis delineates distinctive clusters across regions: the Middle East, Latin America, and China form one cluster (cluster 4) characterized by low HIV burden, average high annual GDP, and high UHC. This suggests the potential to incorporate HIV services into general healthcare systems. Meanwhile, clusters one and two, encompassing nations in sub-Saharan Africa, exhibit variations in epidemic nature—concentrated versus generalized. This distinction signifies opportunities for tailored service delivery strategies. Notably, disparities in clusters one and two may stem from factors like access to care for key populations and the level of service integration. At the same time, our results indicate that, using the currently available indicators, clustering largely mirrors regional groupings adjusted for income and

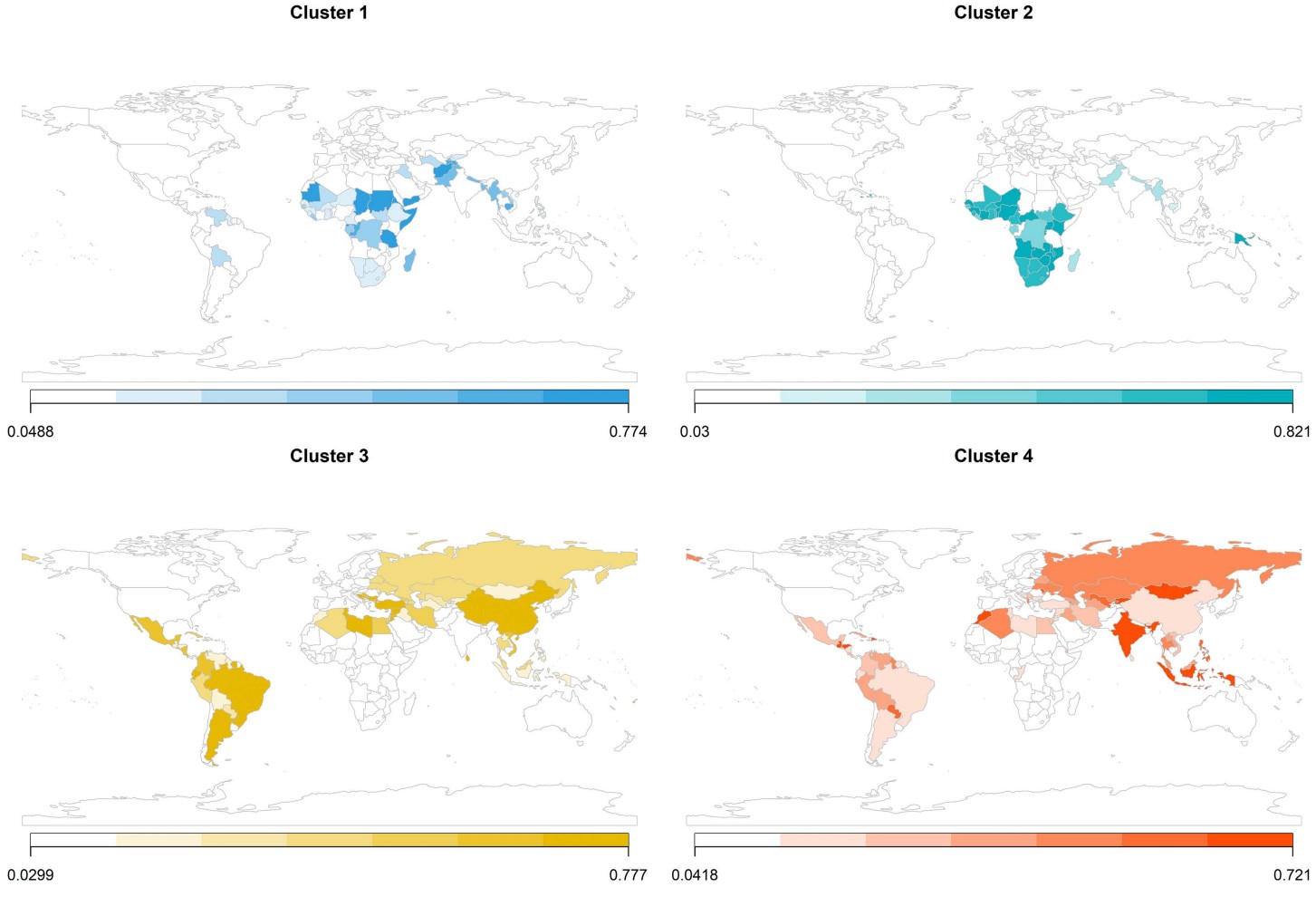

**Fig 4. Country maps for fuzzy Partitioning Around Medoids (FANNY) cluster assignments.** The colour gradient indicates each country's degree of membership (ranging from 0.00 to 1.00) in a given cluster. Higher values represent a stronger association with that cluster, showing that countries can partially belong to multiple clusters rather than being assigned to just one. Source of the basemap shapefile: a SpatialPolygonsDataFrame object containing a simplified world map with polygons attributed to 244 countries, based on Natural Earth data (© Natural Earth, www.naturalearthdata.com), public domain.

epidemiology—factors already considered in policymaking—while also revealing crossovers that offer opportunities for further investigation.

By considering many indicators across relevant domains, we aimed to reduce bias introduced in more traditional analyses by handpicking a few indicators relevant to HIV service delivery and financing. A recent study by Kavanagh *et al.* [29] suggested significant variation exists in the content and adoption of HIV-related laws and policies between countries, regions, and policy areas. Our study adds to their work by considering the relevant HIV dimensions beyond laws and policies (i.e., healthcare financing, HIV/AIDS burden) and considering possible confounders (i.e., demographic and socioeconomic indicators and health and mortality indicators).

Compared to existing UNAIDS and WHO classifications, which often group countries by geographic region or epidemic burden [14,15], our ML-derived clusters highlight cross-regional similarities and divergences. For example, while sub-Saharan Africa largely groups together under both approaches, our analysis unexpectedly clusters China with Latin

**Table 2. Summarized key input variables by identified clusters. Median and interquartile range (IQR) for continuous variables, proportion of countries for binary variables. Cramer's correlation coefficient was calculated between clusters and world region or HIV/AIDS prevalence based on quartiles for continuous variables and directly for binary variables. Imputed data was not used to create this table.**

| | Cluster 1 | Cluster 2 | Cluster 3 | Cluster 4 | Correlation coefficient |
|---|---|---|---|---|---|
| **Cluster profile name** | *Concentrated epidemic; upper-middle income GDP; higher UHC service coverage* | *Generalized HIV epidemic; low- and lower-middle income GDP; lower UHC service coverage* | *Low HIV prevalence; low- and upper-middle income GDP; high UHC service coverage* | *Low HIV prevalence; upper-middle income GDP; medium UHC service coverage* | |
| N of countries | 19 | 38 | 51 | 26 | |
| Regions represented | SSA/EAP | SSA | ECA/LAC/MNA | LAC/EAP/MNA | 0.54 |
| HIV/AIDS prevalence in general population (%) | 0.2 (0.1-1.0) | 2.3 (1.4-4.8) | 0.2 (0.1-0.4) | 0.3 (0.2-0.6) | 0.46 |
| HIV/AIDS prevalence in key population (sex workers) (%) | 4.0 (1.1-8.8) | 14.4 (7.8-36.0) | 1.2 (0.6-3.1.0) | 1.6 (1.0-4.2) | 0.43 |
| HIV/AIDS prevalence in key population (men-who-have-sex-with-men [MSM]) (%) | 8.4 (3.0 – 13.6) | 12.9 (6.7-18.2) | 8.9 (3.7-11.9) | 7.3 (4.8-12.5) | 0.15 |
| **Variables identified by Random Forest classifier (top 20)** | | | | | |
| Int'l HIV vertical transmission prevention ratio | 1.00 (0.96-1.00) | 1.00 (0.96-1.00) | 0.02 (0.00-0.12) | 0.50 (0.38-0.63) | |
| Total public prevention ratio | 0.02 (0.01-0.08) | 0.01 (0-0.08) | 0.77 (0.58-0.89) | 0.21 (0.01-0.48) | |
| Int'l prevention ratio | 1.00 (0.97-1.00) | 0.98 (0.93-1.00) | 0.16 (0.08-0.27) | 0.81 (0.53-1.00) | |
| Life expectancy at birth (years) | 64.66 (63.63-66.49) | 61.01 (58.12-63.6) | 75.04 (72.41-76.31) | 70.80 (68.90-72.42) | |
| Dentists per 10,000 population | 0.50 (0.08-0.81) | 0.09 (0.02-0.20) | 2.95 (1.62-6.58) | 1.36 (0.71-1.85) | |
| Under-5 mortality rate | 58.0 (45.5-66.8) | 75.5 (65.0-99.0) | 17.8 (17.5-20.3) | 35.0 (32.6-40.0) | |
| Total public treatment & care ratio | 0.22 (0.16-0.28) | 0.10 (0.00-0.33) | 0.99 (0.87-1.00) | 0.67 (0.36-0.85) | |
| Population using basic sanitation (%) | 45.93 (31.13-58.64) | 31.92 (17.71-45.14) | 91.49 (86.52-96.41) | 78.31 (64.11-88.56) | |
| Population aged 0–14 (%) | 40.39 (35.81-43.72) | 42.49 (38.71-44.89) | 24.02 (19.03-27.72) | 31.92 (28.96-34.91) | |
| Maternal mortality ratio (per 100,000 live births) | 308 (193-575) | 402 (257-559) | 34 (24-56) | 94 (63-150) | |
| HIV prevalence in general population (%) | 0.2 (0.1-1.0) | 2.3 (1.4-4.8) | 0.2 (0.1-0.4) | 0.3 (0.2-0.6) | |
| Female life expectancy at 60 (years) | 17.53 (17.19-18.04) | 17.30 (15.87-18.15) | 21.90 (20.59-23.00) | 19.91 (18.81-22.12) | |
| Global Fund resources (in million USD) | 17.6 (12.9-35.0) | 40.4 (25.7-161.9) | 57.0 (19.3-102.8) | 8.5 (4.0-13.3) | |
| Physicians per 1,000 population | 0.19 (0.11-0.39) | 0.08 (0.05-0.17) | 1.81 (1.12-2.88) | 0.62 (0.34-1.29) | |
| Int'l treatment & care ratio | 0.92 (0.75-1.00) | 0.82 (0.60-1.00) | 0.02 (0.00-0.17) | 0.33 (0.12-0.68) | |
| Total public HIV vertical prevention ratio | 0.00 (0.00-0.09) | 0.04 (0.00-0.22) | 0.99 (0.88-1.00) | 0.83 (0.51-1.00) | |
| Nurse-initiated ART allowed (yes/no) | no | yes | no | yes | |
| Population aged 65+ (%) | 2.90 (2.61-4.07) | 2.90 (2.72-3.46) | 7.93 (5.9-10.99) | 4.59 (4.16-5.97) | |
| GDP per capita (PPP, current USD) | 3114.90 (2068.75-4410.01) | 2990.28 (1878.81-4726.98) | 13072.13 (11071.50-16237.12) | 6843.56 (4692.55-10494.47) | |

Abbreviations: SSA = Sub-Saharan Africa, LAC = Latin America and the Caribbean, EAP = East Asia and Pacific, ECA = Europe and Central Asia region, MNA = Middle East and North Africa, GDP = Gross Domestic Product, UHC = Universal Health Coverage, USD = United States Dollars, PPP = Purchasing Power Parity, ART = Antiretroviral Therapy

America, and identifies outliers such as Sri Lanka and Papua New Guinea—patterns not visible in standard typologies. This illustrates how ML approaches can supplement existing classification models by dynamically incorporating financing and service delivery dimensions. For instance, Zimbabwe and the Caribbean both fall under different regions and face different local epidemics, but both have extensive community healthcare worker programs and currently learning from each other to further strengthen their local health responses [30].

The added value of ML lies in its capacity to incorporate multi-criteria data—such as legal environments for key populations, integration of HIV services into UHC, and health spending efficiency—which could yield typologies directly applicable to policymaking. In the future, moving towards the cascade targets, countries that have already achieved the 95–95–95 targets [1], such as Botswana and Rwanda, may offer lessons to others within their cluster that face similar system constraints but are at an earlier stage of transition.

Comparable studies that assessed health system performance suggest that unsupervised ML analysis is a helpful tool for identifying typologies through a mix of quantitative and institutional indicators [17,31]. A study on long-term care systems in OECD countries underlines that when systems transform rapidly, previous regional-based typologies become outdated and can be supplemented by ML approaches [17]. Kavanagh et al. demonstrated, through the work with the HIV Policy Lab, that HIV-related law and policy environments vary widely across countries and regions, underscoring the limitations of relying solely on geographic or epidemic classifications [29]. Allel et al. (2022) further showed how efficiency in HIV/AIDS spending differs significantly across 78 countries, highlighting the value of multi-dimensional approaches that integrate financing and service delivery factors [32]. Together, these studies support the use of unsupervised methods to generate typologies that reflect not only epidemiological and regional patterns but also legal, financial, and systemic dimensions of the HIV response.

A key benefit of our approach is that analyses can be completed in days using minimal computing resources and open-source software like R. We provide a reproducible workflow—including data preparation, dimensionality reduction, clustering, and validation—available on GitHub [28]. While expert input is still needed to select relevant indicators and interpret clusters in a policy context, this method greatly accelerates and simplifies the generation of actionable typologies.

Our study has several limitations. The unsupervised ML methods used in this study are explorative; no gold standard exists to validate the clustering process and results. The results from the ML models should be interpreted in light of the chosen indicators, as indicators directly drive the results. To increase usefulness for policy and practice, selected indicators are based on existing dimensions of relevant indicators to (sub)national HIV healthcare delivery and financing strategies as defined by UNAIDS [21]. The selected indicators are comparable to those chosen in other HIV strategic planning studies [32,33]. In addition, we included indicators that reflected the status of UHC, such as health worker density and out-of-pocket spending on health. We could not include some valuable indicators for which no data were currently available, for example, user fees, which are known to impact health outcomes in LMICs and could reveal further insights [34].

Overall, the clusters were relatively robust, though stability varied. Clusters 2 and 3 were highly stable, while cluster 1 was less cohesive. Fuzzy clustering confirmed this pattern, with most countries clearly assigned in one cluster, but a few exhibited mixed affiliations, highlighting areas of overlap and underlying heterogeneity. Although most studies do not explicitly report cluster statistics, we compared our methodology to that of Carrillo-Larco et al. [19], who used five country-level variables (among the UHC service coverage index) to classify countries by COVID-19 status. They considered silhouette scores of 0.42–0.44 acceptable, which is below our cluster quality score (0.53). Altogether, while the data being a mix of demographic, epidemiological and institutional/political indicators adds to the study's strength in terms of completeness, it also makes information less precise and structured for ML analysis. For future analyses, using more granular data inputs, for example about legislation about human rights, or differentiated by sub-national level, ML could generate more actionable typologies, moving beyond broad regional categories to provide a more efficient tool for identifying cross-country innovations and decision-making.

## Conclusions

In this study, we applied ML methods to identify distinctive country profiles of HIV service delivery and financing, providing a "proof of principle" for the use of ML in strategic public health analysis. While our current results largely reproduce regional groupings adjusted for income and epidemiology, they also reveal unexpected cross-regional similarities and outliers not captured by existing UNAIDS or WHO classifications. This study illustrates the potential of ML methods to generate useful insights and to support policy and intervention planning in the global HIV response. The clusters identified here provide a basis for comparative case studies among countries with similar profiles, while stakeholder engagement will be essential to translate these to best policy practices. While exploratory, our analysis demonstrates both the feasibility of ML approaches in global health and the importance of investing in richer datasets to enhance their policy relevance. In the current context of drastic funding reduction, however, country partners and global health stakeholders have a short time to pursue urgent reforms to safeguard the future of this vital program and save millions of lives. Identifying groups of countries with similar entry points and challenges across the five areas could help accelerate identifying innovative models tailored to a country's needs that transcend the traditional regional grouping.

## Supporting information

**S1 Text. Supporting information.** Supporting information for *Using country-level variables to discover country clusters beyond traditional health policy and performance metrics: An unsupervised machine learning approach for HIV healthcare delivery and financing.*
(DOCX)

## Author contributions

**Conceptualization:** Zoïe W. Alexiou, Caroline A. Bulstra, Till Bärnighausen.

**Data curation:** Zoïe W. Alexiou, Siddharth Dixit.

**Formal analysis:** Zoïe W. Alexiou, Siddharth Dixit.

**Funding acquisition:** Caroline A. Bulstra, Till Bärnighausen.

**Investigation:** Siddharth Dixit.

**Methodology:** Zoïe W. Alexiou, Siddharth Dixit, Stefan Kohler, Caroline A. Bulstra.

**Project administration:** Fern Terris-Prestholt, Iris Semini.

**Resources:** Fern Terris-Prestholt, Iris Semini.

**Supervision:** Osondu Ogbuoji, Caroline A. Bulstra.

**Validation:** Osondu Ogbuoji, Stefan Kohler, Rifat Atun, Caroline A. Bulstra, Till Bärnighausen.

**Visualization:** Stefan Kohler, Caroline A. Bulstra.

**Writing – original draft:** Zoïe W. Alexiou, Siddharth Dixit, Caroline A. Bulstra.

**Writing – review & editing:** Osondu Ogbuoji, Stefan Kohler, Rifat Atun, Fern Terris-Prestholt, Iris Semini, Caroline A. Bulstra, Till Bärnighausen.

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
