## [Decision Letter · Decision Letter 0]

18 Jun 2025

PGPH-D-25-00288

Identifying clusters of countries with similar progression towards sustainable HIV service delivery and financing: an unsupervised machine learning approach

Dear Dr. Bulstra,

Thank you for submitting your manuscript to PLOS Global Public Health. After careful consideration, we feel that it has merit but does not fully meet PLOS Global Public Health’s publication criteria as it currently stands. Therefore, we invite you to submit a revised version of the manuscript that addresses the points raised during the review process.

The manuscript has been evaluated by two reviewers, and their comments are available below.

The reviewers have raised a number of major concerns. Could you please carefully revise the manuscript to address all comments raised?

We look forward to receiving your revised manuscript.

Kind regards,

Johanna Pruller, Ph.D.

PLOS Staff Editor

Journal Requirements:

1. Please note that PLOS Global Public Health has specific guidelines on code sharing for submissions in which author-generated code underpins the findings in the manuscript. In these cases, all author-generated code must be made available without restrictions upon publication of the work. Please review our guidelines at https://journals.plos.org/globalpublichealth/s/materials-and-software-sharing#loc-sharing-code and ensure that your code is shared in a way that follows best practice and facilitates reproducibility and reuse.

Additional Editor Comments (if provided):

Reviewers' comments:

Reviewer's Responses to Questions

**Comments to the Author**

1. Does this manuscript meet PLOS Global Public Health’s publication criteria?

Reviewer #1: Yes

Reviewer #2: Yes

2. Has the statistical analysis been performed appropriately and rigorously?

Reviewer #1: Yes

Reviewer #2: Yes

3. Have the authors made all data underlying the findings in their manuscript fully available (please refer to the Data Availability Statement at the start of the manuscript PDF file)?

Reviewer #1: Yes

Reviewer #2: Yes

4. Is the manuscript presented in an intelligible fashion and written in standard English?

Reviewer #1: Yes

Reviewer #2: Yes

Reviewer #1: 1. Why chose K-means clustering?

2. Why chose PCA? Scree plot indicates that the variables are not in relative high correlation, then what the purpose of PCA? PCA could additionally reduce the interpretation, though.

3. 25% of the total variance explained is too low for PCA even though authors claimed their purpose was to highlight groups. At least a sensitivity analysis of using more PCs that count for more variance (e.g. 80%) is needed.

4. How can you know your clusters are (not) mainly determined by general socioeconomic factors or general health level instead of HIV related factors. For example, one could just use K-means clustering general socioeconomic factors that results in similar 4 clusters, and post hoc compare some HIV indicators.

5. Figure 1A needs more explanation. Readers may not understand the meaning of it.

Reviewer #2: Major Concerns

1-Over-reliance on K-means clustering

Although the authors mention K-means, hierarchical clustering, and Fuzzy K-means, the core analysis depends solely on K-means. This approach is sensitive to initial centroid selection and Euclidean distance in high-dimensional data. The study lacks comparative experiments (e.g., DBSCAN, Gaussian Mixture Models, Spectral Clustering) to improve robustness.

2-Insufficient variance explained by PCA

The authors retain only three principal components (explaining ~25% of variance), which is unusually low for social science research. While they justify this as "highlighting groups," such low information retention may distort clustering. They should explore retaining more components or using nonlinear methods (e.g., t-SNE, UMAP).

3-No stability or statistical significance tests

Beyond Silhouette and Adjusted Rand Index, the study does not assess cluster stability (e.g., bootstrapping, Cluster Stability Analysis, Gap Statistic). Without evaluating robustness to sample variations, the reliability and reproducibility of results remain unclear.

4-Lack of empirical or policy validation

The authors claim policy relevance but do not apply clusters to real-world HIV program comparisons or link them to independent data (e.g., health expenditure, outbreak control). The work remains at a "proof-of-concept" stage with weak policy connections.

5-Unclear handling of high-dimensional heterogeneous data

The study integrates 120 diverse variables (institutional, binary, continuous) using FAMD but lacks transparency in weighting, scaling, and outlier control. This may affect clustering quality and fairness.

Minor Comments

1-Missing data and imputation accuracy unreported

Provide missing rates per variable category and evaluate imputation errors (e.g., RMSE or comparison with complete cases) to avoid bias.

2-Insufficient analysis of Fuzzy Clustering results

The fuzzy clustering map is provided, but membership probabilities and mixed-boundary countries are not discussed in depth.

3-No external or temporal validation

All data are post-2017. Consider a hold-out validation set or temporal splits to test generalizability.

4-Undefined evaluation thresholds

The Silhouette score (0.35) lacks a reference standard (e.g., >0.5 for good clustering). Clarify thresholds and discuss low-confidence clusters.

5-No feature importance analysis

Unsupervised feature importance (e.g., Random Forest-based) could identify key variables for policy interpretation.

6-Limited comparison with prior work

The discussion does not benchmark against existing WHO/UNAIDS classifications or models (e.g., Kavanagh et al.), reducing persuasiveness.

7-Unaddressed within-cluster heterogeneity

Countries like "South Africa vs. Namibia" or "China vs. Mexico" may differ significantly in policy and epidemic management. Explain their grouping and potential misclustering.

**Do you want your identity to be public for this peer review?** For information about this choice, including consent withdrawal, please see our Privacy Policy

Reviewer #1: No

Reviewer #2: No

---

## [Decision Letter · Decision Letter 1]

30 Oct 2025

Using country-level variables to discover country clusters beyond traditional health policy and performance metrics: An unsupervised machine learning approach for HIV healthcare delivery and financing

PGPH-D-25-00288R1

Dear Dr. Bulstra,

We are pleased to inform you that your manuscript 'Using country-level variables to discover country clusters beyond traditional health policy and performance metrics: An unsupervised machine learning approach for HIV healthcare delivery and financing' has been provisionally accepted for publication in PLOS Global Public Health.

Best regards,

Julia Robinson

Executive Editor

Reviewer Comments (if any, and for reference):

Reviewer's Responses to Questions

**Comments to the Author**

Reviewer #1: All comments have been addressed

publication criteria?

Reviewer #1: (No Response)

3. Has the statistical analysis been performed appropriately and rigorously?

Reviewer #1: (No Response)

4. Have the authors made all data underlying the findings in their manuscript fully available (please refer to the Data Availability Statement at the start of the manuscript PDF file)?

Reviewer #1: (No Response)

5. Is the manuscript presented in an intelligible fashion and written in standard English?

Reviewer #1: (No Response)

Reviewer #1: (No Response)

**Do you want your identity to be public for this peer review?** For information about this choice, including consent withdrawal, please see our Privacy Policy

Reviewer #1: No
